# Acceptability of surgical care in Uganda: a qualitative study on users and providers

Paula Rauschendorf [1], Rosette Nume,[2] Walter Bruchhausen[1]

¹Section Global Health, Institute of Hygiene and Public Health, University of Bonn, Bonn, Germany
²School of Women and Gender Studies, Makerere University, Kampala, Uganda

**Correspondence to**
Dr Paula Rauschendorf;
p.rauschendorf@outlook.de

## ABSTRACT

**Objectives** This study was conducted to assess acceptability of surgical care in Eastern Uganda and enable better allocation of resources, and to guide health policy towards increased surgical care seeking.

**Design** This qualitative study used semistructured in-depth interviews that were transcribed and analysed by coding according to grounded theory.

**Setting** The study was set in Eastern Uganda in the districts of Jinja, Mayuge, Kamuli, Iganga, Luuka, Buikwe and Buvuma.

**Participants** Interviews were conducted with 32 past surgical patients, 16 community members who had not undergone surgery, 17 healthcare professionals involved in surgical treatment and 7 district health officers or their deputies.

**Results** The five intersecting categories that emerged were health literacy, perceptions, risks and fears, search for alternatives, care/treatment and trust in healthcare workers. It was also demonstrated that considering the user and provider side at the same time is very useful for a more extensive understanding of surgical care-seeking behaviour and the impact of user–provider interactions or lack thereof.

**Conclusion** While affordability and accessibility are well defined and therefore easier to assess, acceptability is a much less quantifiable concept. This study breaks it down into tangible concepts in the form of five categories, which provide guidance for future interventions targeting acceptability of surgical care. We also demonstrated that multiple perspectives are beneficial to understanding the multifactorial nature of healthcare seeking and provision.

## STRENGTHS AND LIMITATIONS OF THIS STUDY

⇒ This study provides multiple perspectives by analysing interviews with local health officials, (potential) users, and providers of surgical care.
⇒ Among the interviewees of each group, diversity was created through purposive sampling.
⇒ This study is limited by its small sample size and geographical area, despite its attempt for representativity.
⇒ After analysis of the first set of interviews, some questions were rephrased following grounded theory to increase clarity.

## INTRODUCTION

In 2015, the Lancet Commission Report on Global Surgery highlighted the inequity of the global burden of surgical disease and access to care.[1] Surgical care, which is defined as 'any intervention directed at reducing the disability or premature death associated with a surgical condition',[2] has been found to be highly effective in reducing the burden of disease and disability-adjusted life-years lost.[3] Uganda faces a high unmet need for surgery,[4–6] with few operating theatres (OTs) (0.2 major OTs per 100 000 people)[7] and a low surgeon density of 0.73 surgeons per 100 000 persons.[6] Wards are overbooked[8] and shortages of necessities such as water, electricity, equipment and limited access to blood banks further hinder safe surgical care.[9]

The inequity in the burden of surgical disease and access to care, in the context of Universal Health Coverage as a target of the Sustainable Development Goal 3[10] underscores the importance of accessible, affordable, and acceptable surgical care. While accessibility[11–13] and affordability[14] of surgical care have been thoroughly researched in Uganda, the acceptability of surgical care has not been a focus.

Acceptability of care was defined by Sekhon *et al* as 'a multifaceted construct that reflects the extent to which people delivering or receiving a healthcare intervention consider it to be appropriate, based on anticipated or experienced cognitive and emotional responses to the intervention.'[15] It is influenced by a multitude of factors such as the quality of healthcare facilities, availability and attitudes of and towards healthcare personnel, perceptions, risks, and fears associated with healthcare, available alternatives, and health literacy, which can be defined as 'the degree to which individuals have the ability to find, understand, and use information and services to inform health-related decisions and actions for themselves and others.'[16]

Research on acceptability of care can provide insights into reasons for low uptake of surgical care and inform resource allocation and health policies. Acceptability of care needs to be assessed from multiple perspectives: patients, the broader public, healthcare providers, and health officials can each provide a unique viewpoint. Therefore, this study analyses interviews with former surgical patients (FP), community members (CM), surgical healthcare workers (HW) and district health officers (DHO).

## METHODS

### Study design

This qualitative study used semistructured in-depth interviews that were transcribed, translated if needed, and analysed by coding according to grounded theory. Grounded theory is a qualitative research method aimed at developing theories from data and their analysis[17–19] that can increase understanding of social behaviours and processes, for example, in patient care.[20] Due to its inductive nature, it is often used in fields with little prior research.

### Patient and public involvement

There was no patient or public involvement in the study prior to data collection.

### Setting and participants

The interviews were conducted from August to September 2019 in the districts of Jinja, Mayuge, Kamuli, Iganga, Luuka, Buikwe, and Buvuma. They are located in South-Eastern Uganda and have an estimated population of 2.84 million.[21] The Eastern region of Uganda has a significantly higher poverty rate than the national average.[22]

We interviewed members of four groups in each district:

- ▶ Six FPs and
- ▶ Three CMs who had not had surgery from three different communities which were selected purposively with regards to distance to the next surgery providing healthcare facility and distance from the main road. One community was chosen near a surgical facility, one was chosen to be further from a facility but along the main road, and one was chosen further away from both the facility and the main road.
- ▶ Three HW involved in surgical care, including representatives from a private for-profit (PFP), private not-for-profit (PNFP), and government (public) facility. In districts where PFP or PNFP facilities were lacking, either two interviews were conducted at a government facility, or one interview was conducted in the district. To ensure diversity and minimise selection bias, different medical cadres were interviewed in each sector, following purposive sampling.
- ▶ The DHO.

The sampling strategy is provided in figure 1.

### Interview guide

The process of developing the interview guides involved reviewing relevant literature, identifying key concepts or themes, and crafting open-ended questions to elicit in-depth responses from the interviewees. The final interview guides were structured in a logical sequence to facilitate a smooth flow of the interviews and to ensure that all relevant topics were covered. The interview guides served as a flexible tool that allowed for probing and follow-up questions based on the responses of the interviewees, ensuring that a rich and comprehensive data set was collected during the interviews. After analysis of the first set of interviews, some questions were rephrased for clarity.

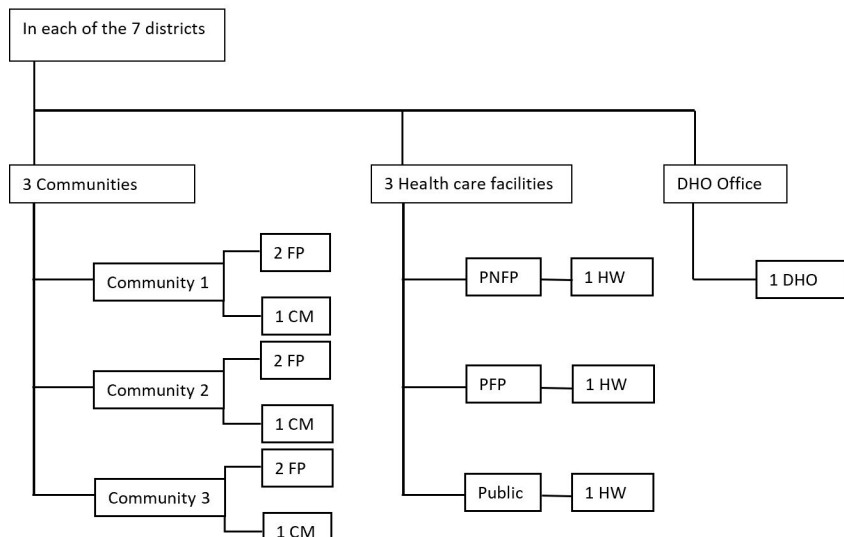

**Figure 1** Sampling strategy per district. The goal was to conduct six interviews with former patients (FP) and three interviews with community members (CM) from three different communities, three interviews with healthcare workers (HW) from distinct types of facilities (one from a private for-profit (PFP) facility, one from a private not-for-profit (PNFP) facility, and one from a public facility), and one interview with the District Health Officer (DHO).

During the interviews, FPs were asked about their experience with surgery, delays in healthcare seeking, reasons for uncertainty, consultations outside of biomedical healthcare, decision making, the process of their surgical treatment, changes in perception of surgery, community support, and suggestions for improvement of health facilities. CMs were asked about their thoughts on surgery and perceived struggles, their health behaviours, knowledge of surgical conditions, financial impact, health insurance coverage, and their concerns pertaining to surgery.

HWs were asked about their experience in providing surgical care, difficulties, improvements and patient perceptions. Interviews with DHOs commenced with a question about obstacles in providing access to healthcare, particularly surgical care, followed by questions about supply chains, health education campaigns, the surgical workforce, and involvement of non-governmental organisations .

For full interview guides please refer to online supplemental appendix 1–4.

### Data collection

Overall, 72 interviews were conducted. Community interviews were conducted in Lusoga. Interviews with HWs and DHOs were conducted in English or Lusoga depending on preference.

The DHO was approached for an interview and simultaneously permission was obtained to interview HWs in the district. Then HWs were invited to participate in the study at their respective healthcare facilities. CMs were engaged and asked if they had undergone surgery or could refer to someone who had. This process was repeated until two FPs and one CM were recruited.

Theoretical saturation was reached towards the end of sampling, defined as the 'point in category development at which no new properties, dimensions or relationships emerge during analysis'[23] and 'new data collection (indicated by theoretical sampling) does not lead to any further changes to the theory'.[24] This suggests that the sample size was sufficient for addressing the research questions and achieving the research objectives.

Interviews were audio recorded, transcribed verbatim, and anonymised. Translations to English were performed if the original language of the interview was not English.

### Data analysis

Data were analysed using MAXQDA Pro Analytics. MAXQDA is a qualitative data analysis software that provides various tools and features for analysing qualitative data. The software allows researchers to organise, code and analyse text, audio, video, and image data in a systematic and rigorous manner. Interview transcripts were coded according to grounded theory by finding the keywords in each section. Codes were grouped into subcategories and sorted into larger categories.

## RESULTS

Interviews were conducted with 33 FPs, 16 CMs, 17 HWs, and 7 DHOs or their deputies.

Demographic data of the sample can be found in tables 1 and 2.

During the analysis, five dimensions of acceptability of surgical care emerged. Health literacy was found to impact healthcare-seeking behaviour and shape expectations about surgical care, which were further influenced by fears and perceptions of surgery. Comparisons were made between surgical care and alternative healthcare options. Trust in the healthcare worker, as well as care and treatment, influenced patients' experiences and attitudes.

### Health literacy: impact and improvement

Low levels of health literacy impacted healthcare-seeking negatively according to many participants from all groups. A DHO lamented that patients did not know the '*timely need for coming for healthcare*'. HWs supported this and explained that '*people don't know that they require surgery*' (surgeon, PNFP) or '*the impact of their condition*' (medical doctor (MD), PNFP). An FP mentioned that a mass in his spleen was discovered during an intestinal obstruction surgery one year before, but he did not seek further medical evaluation afterward.

Many FPs reported waiting for a long time before seeking care, as one said: '*I delayed a bit. I think it was around […] 10 years*' (FP, PFP). Along with other participants, another FP attributed the delay to his '*ignorance, because I wasn't aware the illness needed a medical treatment*' (FP). Some participants described self-medicating, as in this example: '*When I am sick, I just buy medicine of 2000 to 3000 shillings and if that sickness doesn't respond to the treatment, I continue changing the type of treatment until I get better.*' (CM)

Low levels of general education were discussed, since '*perceptions are about education*' (medical officer (MO), public). An HW described struggling to find a comprehensible way to explain surgical conditions to the patients. CMs similarly said: '*after taking blood samples from us, there's nothing you can understand*' (CM) and '*those who never went to school don't know anything and you have to believe all what the healthcare workers tell you*' (CM).

While many CMs demonstrated knowledge of obstetric conditions and hernia, other conditions such as goitre, ulcer and cancer were less frequently mentioned. An HW attributed this to the fact that '*surgery doesn't have enough advocacy, but obstetric, emergency obstetric care has received a lot of advocacy and a lot of funding*' (MO, public). He described the increase in health education campaigns around obstetric emergencies had led to women '*coming to the facilities, seeking care, surgical care compared to general surgery.*' (MO, public). Another MO (PNFP) confirmed the importance of such campaigns for recruiting patients.

Therefore, more health education specific to surgery was proposed to improve surgical care seeking. Suggestions included campaigns broadcasted via radio and

**Table 1** Demographic characteristics of FP and CM

| Characteristic | FPs | Percentage | CMs | Percentage |
|---|---|---|---|---|
| Sex | | | | |
| Female | 21 | 63.6 | 10 | 62.5 |
| Male | 12 | 36.4 | 6 | 37.5 |
| Age (years) | | | | |
| <20 | 3 | 9.1 | 0 | 0.0 |
| 21–40 | 12 | 36.4 | 3 | 18.8 |
| 41–60 | 12 | 36.4 | 10 | 62.5 |
| 61–80 | 5 | 15.2 | 3 | 18.8 |
| >80 | 1 | 3.0 | 0 | 0.0 |
| Level of education | | | | |
| Tertiary | 3 | 9.1 | 2 | 12.5 |
| Secondary | 8 | 24.2 | 3 | 18.8 |
| Primary | 9 | 27.3 | 11 | 68.8 |
| None | 5 | 15.2 | 0 | 0.0 |
| No answer | 8 | 24.2 | 0 | 0.0 |
| Employment status | | | | |
| Self-employed | 2 | 6.1 | 4 | 25.0 |
| Unemployed | 16 | 48.5 | 11 | 68.8 |
| Formal employment | 6 | 18.2 | 0 | 0.0 |
| Retired | 1 | 3.0 | 0 | 0.0 |
| No answer | 8 | 24.2 | 1 | 6.3 |
| Time of surgery | | | | |
| 2015–2019 | 15 | 45.5 | | |
| 2010–2014 | 4 | 12.1 | | |
| 2005–2009 | 3 | 9.1 | | |
| Before 2005 | 1 | 3.0 | | |
| No answer | 10 | 30.3 | | |
| Facility type | | | | |
| Public | 16 | 48.5 | | |
| PNFP | 5 | 15.2 | | |
| PFP | 8 | 24.2 | | |
| No answer | 4 | 12.1 | | |

CM, community member; FP, former patient; PFP, private for profit; PNFP, private non-for-profit.

television as well as outreaches and village health teams (VHTs) '*to identify some of these complications—the illnesses—and educate, give some health education and refer these people to where they can get services*' (Surgeon, PFP). One HW raised concerns that educating the population had to be linked with building surgical capacity to accommodate the anticipated increase in surgical patients.

### Perceptions and fears of surgery

Surgery was perceived by CMs and FPs as a way of '*reducing the sickness that is bothering you*' (FP, PNFP) by '*cutting you open to see the cause of sickness and remove the unwanted stuff*' (FP, PFP), oftentimes with urgency: '*the patient is in a critical and not understandable condition, then why can that patient not be taken to be operated*' (CM). Surgical care was described as a last resort by CMs and FPs, often only an option '*when the condition that calls for the operation is very extremely severe, when the pain is so so so beyond, without giving you time to eat, drink or walk, I mean when the illness is at its maximum, then there you can decide to go for the operation*' (CM).

The risk associated with surgery was often named as '*you can come back or not come back*' (CM). A surgeon (PNFP)

**Table 2** Demographic characteristics of HW

| Characteristic | HWs | Percentage |
|---|---|---|
| Sex | | |
| Female | 4 | 23.5 |
| Male | 13 | 76.5 |
| Profession | | |
| Nurse | 1 | 5.9 |
| Midwife | 1 | 5.9 |
| Theatre assistant | 1 | 5.9 |
| Medical officer | 4 | 23.5 |
| Medical doctor | 6 | 35.3 |
| Surgeon | 3 | 17.6 |
| Obstetrician | 1 | 5.9 |
| Facility type | | |
| Public | 8 | 47.1 |
| PNFP | 5 | 29.4 |
| PFP | 4 | 23.5 |

HW, healthcare worker; PFP, private for profit; PNFP, private non-for-profit.

attributed these sceptical attitudes to the fact that '*in the past, the results from surgery have not been good. It has been between life and death.*' One of the DHOs mentioned that cases with complications are overly emphasised causing a prevalent misconception that surgery is unsafe. Seeming medical malpractice in anaesthesia was described by one FP (PFP): '*When I was still half sedated, they cut me but the way they pulled my scrotum with much force I felt terrible pain in the intestines*'.

Participants mentioned the fear that surgeons would leave instruments or swabs in the patient with possible complications and reoperations. One recounted that the people '*normally say that when they go for surgery, they make you more disabled, not allowed to do certain work*' (CM). Another example was the fear that putting a patient '*on oxygen means he's going to die*' (nurse, PNFP). Also, there were '*rumours that healthcare workers steal certain organs from the patients*' (CM) which led to fear in the community. One FP (public) suggested having a relative of the patient in the theatre to '*see the procedures of the operation and what they have removed from the patient*'.

One midwife (PFP) saw an increase in trust after successful operations, which was also apparent in some interviews with FPs: '*healthcare providers can operate and after the operation, you can wake up and resume with normal walking, healthy person, and the illness is taken away you can become better, and you live a healthy life.*' (FP, PNFP) Another FP (PFP) described how she was asked after her surgery '*not to scare my fellow women that the health facility was a terrible place; they told me to advise my friends to go for medical procedures*'. This demonstrates that staff feels the need for improving the reputation of surgical treatment. As a way

of counteracting patient fear, effective preoperative counselling was suggested by FPs and HWs.

## Understanding of illness and the search for alternatives

Surgical conditions as seen from a biomedical perspective were mostly regarded by FPs and CMs as necessitating biomedical treatment, and many participants named biomedical facilities as their only source of healthcare. Patients mostly became aware that their condition was surgical after it had been identified as such by an HW. One FP (PNFP) stated: '*sickness that is meant for operation, still goes for operation, there is no alternative*'. In contrast to the view that surgical conditions always necessitate surgery, one medical officer described that pertaining to surgical diagnosis '*they [the patients] think there is something that you have not done. They think you should have given some medication for that condition.*'

Herbal medicine, for example, was described as a way of preventing surgery under certain circumstances: '*I was to have a normal delivery if I used the local herbs*' (FP, public). It was only mentioned in the context of chronic conditions like hernia, not as an alternative in emergency or trauma cases. An FP described '*someone went to those healers and was given herbs and his hernia disappeared, so I also went there and used those herbs for one day but when I noticed no improvement, I gave up on them.*' (FP, PFP). Another example was given by an FP (PFP) with haemorrhoids, who was later operated: '*when I applied the herbs, the piles went back but after some time, it re-occurred, and when I asked them about that, they told me that, the piles can only be healed with traditional medicine and so I continued with that medicine.*' HWs expressed frustration but blamed it on the '*traditional healers or herbalists who keep misguiding or giving false information to our clients*' (surgeon, PFP) and the lack of knowledge concerning surgical conditions.

Similarly, when asked about traditional healers in the area, many respondents voiced distrust because they would not '*reveal that your condition needs surgical treatment. When they want money, they just divert you from seeking medical care, yet they can't treat the problem at hand.*' (CM) However, some participants also saw fault with the patients that sometimes '*cannot differentiate a medical problem from a traditional one*' (CM) and '*think that they have been bewitched, even if the sickness is not traditional*' (FP). This shows an understanding of ill-health and healing of two separate spheres with different illnesses and their corresponding healthcare, that is, an alternative between 'medical' and 'traditional'.

Differing from this understanding of two completely distinct types of illness and treatment, a CM argued '*those other things of bewitching can come later after going to the health facility*' (CM). Here, the common view of complementary functions is expressed that biomedicine may—even quickly—remove the damage like hernia, but not the underlying cause like bewitchment, as one said: '*Can a healthcare worker tell you issues of the home? […] But it is the traditional healer to tell you the facts. The healthcare worker can suspect the cause of illness and tell the patient that maybe the*

*patient delayed getting treatment, but when you go to these ones, they can tell you.'* (CM).

## Care and treatment

The received care was commonly described as good. Good was, however, defined differently by FPs, for example, as not being mistreated or being spoken to '*in a friendly manner'* (FP, PNFP). Others voiced more specific expectations: '*to be cared for, is when the healthcare worker comes in the ward and greets you asking about how the night was, how you feel, the condition of the baby and other issues like that.'* (FP, public) For others, the focus was more on the medical aspects like receiving '*timely treatment'* (FP, PFP).

The type of care received was commonly attributed to the patient's financial situation, where paying the asked amount was linked to good care and medical treatment, whereas patients with financial constraints were often mentioned to receive worse treatment. Other FPs attributed the good care they received to personal relationships with the HWs. One FP (PNFP) reported that '*they respected me because first of all I am educated, and I interacted with them in English during all the chats'*.

Despite general satisfaction, some complaints were voiced like going through episodes of extreme pain after surgery, some to the extent, that it would prevent future surgical care seeking. Other FPs mentioned mistreatment by HWs, for example, a female FP (public) was on a trolley to the operation theatre for a C-section when she '*felt something like a baby coming out then I widened my legs as if I was going to deliver, then the nurse slapped me for widening the legs, and exposing my nakedness to the people'*. The nurse was held accountable for her actions. A CM lamented: '*if you are lucky enough and your patient is operated on, they carelessly stitch and just dump him/her there for you. They are always not concerned with the aftereffects of the operation, it's you to care for the patient.'*

This alludes to the important role of informal caregivers in Ugandan healthcare. '*They carry the burden, the financial, emotional, and so on'* (surgeon, public). Their importance was highlighted multiple times throughout interviews, for example when one FP (public) said, '*if you have no effective caregiver, it's a big challenge'*. The informal caregiver has different responsibilities, mainly taking care of the basic needs of patients while at the hospital and providing emotional support. One FP (public) stated, that when patients did not have their treatment readily prepared by their informal caregiver, nurses '*just pass to another patient'*.

The '*lack of proper trained surgical nursing care'* (medical superintendent) led to some surgeons feeling '*let down by the postoperative care'* (MO, public). A nurse (PNFP) explained: '*we are like 2 or 3 on duty and patients are many and we have to attend to them, sometimes you find that you have not reached up to that peak of caring for the patient'*. Therefore, sometimes '*responsibilities channel to the relatives'* (MO, public) that should be carried out by medical personnel. A surgeon noted that this led to problems in the aftercare of operations. HWs also highlighted that the lack

of separate surgical wards was a challenge to providing specialised care. To counteract these challenges, more training, a separate surgical ward, and an increase in staff were suggested.

Follow-up of operations was another theme that emerged. Some FPs said they had not been asked back for follow-up. HWs commonly said that review dates were scheduled with patients, but attendance varied. This was a challenge for surgical staff, as doctors were then '*not able to monitor them for any complications'* (Surgeon, PFP). Varying commitment from HWs was apparent, with some recording patients' numbers and calling to check-up and some visiting their patients at home. A surgeon (PFP) suggested that regular '*field visits'* could improve follow-up.

## Healthcare workers: trust and misbehaviour

Closely linked to care and treatment was the patient–HW relationship. As one CM stated: '*the healthcare worker is the overall in this thing, he/she is the key to everything'*. Trust and lack thereof were commonly mentioned.

Most interviewed HWs agreed that they felt trusted by patients. Some FPs described a trustful relationship, with one mentioning being taken care of by a doctor off-duty and another calling the doctor a '*saviour'* (FP, public). One medical officer (public) differentiated: '*Those who come, trust us. What they don't trust is the system. What they don't trust is their government'*. Contrastingly, distrust in the surgical skills of healthcare workers was also mentioned with one FP saying: '*most of the healthcare workers are young boys, we don't trust them'*. This is caused by the regulation that demands service in remote communities in the first years after graduation. One doctor (PFP) highlighted that this was a major difference to traditional healers, as they '*tend to give more time to their patients actually and convince them adequately'*. The overwhelming workload and little time spent with the patient were given as reasons for the lack of trust in the healthcare system and HWs.

HWs were expected to '*first think about people's health, then think about money'* (CM) because '*there are rewards in heaven'* (CM). Work ethics of HWs were a concern to participants from the community, who described nurses and doctors commonly asked for informal payments. One FP (public) described being asked '*to pay her 60 000 shillings for quick treatment and we go home instead of being admitted in the ward'*). Some mentioned, HWs would let the patient die if they could not pay: '*There was that first one I saw, he had even refused to operate me, and if the second one never showed up, I was going to die'* (FP, public). Participants suggested to raise salaries of HWs, so they would stop demanding informal payments.

Medical personnel generally described their own work ethics as high, with one medical officer calling it a '*commitment to serve'*. One doctor (PNFP) described having to pay for medical equipment when the patient cannot afford it: '*sometimes it is necessary, where you see that life is more important'*. Nevertheless, many saw issues with motivation and work morale in the health workforce in

general, attributing it to frustration due to poor salaries, long working hours, poorly equipped facilities, and little opportunity for continued education and mentorship. Scholarships for further education and mentorship were mentioned as important. Training programmes through foreign exchange or as part of surgical camps were seen as beneficial for both sides: '*Probably you might come with the skills where you come from, but the context differs. Then you also have something to learn with us, from us working with limited resources.*' (DHO)

## DISCUSSION

In providing surgical care to communities, such as in Uganda, factors beyond accessibility and affordability necessitate careful consideration. While some commonly used models such as the behavioural model of health service use by Andersen[25] consider mostly the user side, this study aims to showcase the user, the provider and the health officials' perspective. The five areas found to impact acceptability of surgical care were: health literacy, perceptions of surgery, the search for alternatives, care and treatment and the relationship to the healthcare worker. These categories, however, had some overlap.

Higher health literacy has been shown to increase acceptance of surgical care[26] while limited health literacy has been linked to worse outcomes and lower satisfaction[27] and to overall non-adherence to preoperative and postoperative instructions.[28] Participants of our study, especially on the provider side, also viewed health literacy on surgical conditions as impactful for improving surgical care seeking. This should be a focus of future interventions, as suggested by Butler *et al* in a study where lack of perceived need left 25% of children with surgical needs untreated.[29] Health literacy assessments have been shown to be feasible and to enable increased patient satisfaction.[30] General education and literacy were also discussed by participants as enabling patients to make better decisions on healthcare seeking for surgical conditions, which was also found by Fuller *et al*[31] and Davé *et al*.[32] Both studies were based on the Surgeons Over Seas Assessment of Surgical Need, a cluster randomised, cross-sectional, national survey employing quantitative methods to assess the prevalence of sugical conditions and related healthcare seeking behaviour.

A thorough understanding of the perceptions of surgery and concepts of illness is necessary to provide acceptable care. The question of what is at stake for those people most concerned by a surgical condition needs to be addressed for meeting important expectations and creating trust. Understandably sceptical attitudes due to former negative experiences with formal education and biomedical healthcare[33] as well as unfavourable surgical outcomes[34] have a large impact on surgical healthcare seeking.

While the surgical treatment was often described as successful by patients, negative perceptions were commonly mentioned. Risks and fears related to surgical interventions act as a barrier for their uptake. Perceptions were also shaped by rumours, like concerns of organ stealing. Rumours like this have existed in many African societies and have been discussed extensively by White[35] and Scheper-Hughes.[36] Other rumours noted were specific to surgical care and the proceedings of an operating room. Improving health literacy, setting a focus on communication with patients and improved education prior to surgery could play an important part in counteracting rumours. It could also reduce self-medication and change the current role of surgery as a last resort or for emergencies only.

With 4.0% of rural children with surgical conditions not receiving care due to fear,[37] trust and positive interactions with surgical care providers are important. While most healthcare personnel felt trusted by patients and many patients voiced trust, some scepticism towards surgical skills particularly of medical officers was expressed. Similar concerns about seniority and skill level were voiced by participants in a study by Mwaka *et al* suggesting that only senior and skilled doctors were regarded as fit to successfully operate on cancer patients.[38] Due to a lack of specialists, non-specialised personnel commonly conducts surgery in Uganda.[39]

Participants in our study, especially medical officers, wished for more mentorship and further specialisation. This would also provide an opportunity for more training on interpersonal skills to improve communication with patients, as suggested by Bohren *et al*.[40] Surgical camps were suggested as possible training opportunities. Modern technology was also proposed to improve mentorship, where medical officers in remote areas would be able to consult with colleagues in Uganda or abroad via telecommunication.

Most FPs and CMs recognised the need for operative care in case of surgical conditions. These findings align with results from a previous study investigating surgery beliefs of lay people, where surgery was seen as 'the appropriate healing method', although there was also 'fear of negative outcome of surgery'.[38] In our study, some participants mentioned attempting alternative treatments under the category of 'traditional medicine', which was also a common theme among providers. Participants from communities often agreed there were indications for traditional medicine, however rarely including surgical conditions such as hernia.

Many participants from all groups highlighted the dangers of misdiagnosis and delay of treatment, which has been previously reported[5 29] and was discussed by WHO in their traditional medicine strategy.[41] However, patients continue to seek care from traditional healers, which healthcare providers attributed to the fact that traditional healers often spend more time with patients. This could be due to the ratio of 1 traditional healer per population of 500, compared with 1 medical doctor per 25 000 people in Uganda.[42] Participants also discussed their role as holistic healers. Some of these features could be a starting point for developing a healthcare system

encompassing the underlying needs of patients answered by traditional medicine.

Overall, patients in this study were satisfied with the received care but noted that factors such as financial situation, education level, and relationships with healthcare workers influenced the type of care received. Similar findings were reported in other studies, citing bribes, socioeconomic class, political affiliation and tribal favouritism[43] as well as social capital and financial leveraging[44] as factors impacting care. Participants in our study linked these behaviours to the work ethics of HWs. While HWs expressed high motivation, many described low overall morale among the health workforce, due to factors such as low salaries, difficult work environments, and lack of mentorship.

Participants described negative interactions between healthcare providers and patients, including harsh treatment and neglect. This is consistent with previous findings from Benin.[34] Kagoya et al found a lack of negotiating power among patients at the Ugandan National Referral Hospital caused by 'illiteracy, language barriers, low socioeconomic status, feeling inferior to HWs and perceived favour to access free services'.[43]

Some FPs described situations that could be interpreted as malpractice or errors in surgical care. A study on general medical errors reported how they occurred and that healthcare professionals were mostly aware of them.[45] In regards to surgical errors, 'inappropriate preoperative management, omitting preoperative investigations and omitting postoperative notes'[46] were most commonly found by reviewing patient records. A lack of error reporting on medication errors was also found by Kiguba et al.[47]

Postsurgical care and follow-up were seen as major challenges by patients and healthcare providers. Lack of trained nursing staff leads to task shifting to informal caregivers, as highlighted by participants. In postoperative caregiving, informal caregivers often lack the necessary training, education, and understanding of medical conditions necessary to provide the best care possible. Formerly proposed caregiver training to increase capacity of postsurgical care[48] was not suggested by participants in this study, who laid more focus on increasing the surgical workforce.

Previous research suggests that receiving follow-up care is strongly dependent on socioeconomic status and distance to access points.[48] Our findings support this. Healthcare professionals in our study suggested taking phone numbers of patients and making fixed appointments to combat this problem. Home visits were mentioned as ideal, but not feasible. A study on maternal and newborn care practices also suggests that a combination of home visits by VHTs and mobile phone consultations improves care.[49] Interventions linked to VHTs should be strengthened and possibly expanded to include regular basic health check-ups to facilitate earlier interventions on complications and remissions.

## LIMITATIONS

In its aim to provide a broad overview of perspectives on surgical care, this study is limited by its small sample size and geographical area, despite its attempt for representativity. However, the broader categories that emerged may apply to the country and could even find application beyond. The study also showed that grounded theory can be applied to study acceptability of care in specific contexts.

In the first interviews, participants were not asked to provide the year of their surgery. In the selection of participants, no exclusion criteria were defined regarding the year of surgery. While this approach may have limited representativity for current surgical care, it was intentional as it was believed that experiences from the past may still shape perceptions and attitudes within the communities, even if the surgery occurred long ago.

The educational, social science, medical and cultural backgrounds of the researchers conducting the interviews, being a Ugandan graduate student and a German medical doctoral student, may have created a sense of authority or power dynamic that could have impacted the participants' answers. This introduces the potential for observer bias in the study, as the presence of the researchers during the interviews could have influenced the responses of the participants.

Concerning both, reliability and validity, some questions needed rephrasing after grounded theory analysis of the first set of interviews since they had been misunderstood.

## CONCLUSION

This study emphasises the need to consider the views of users, providers, and local health officials for a comprehensive understanding of factors influencing the acceptability of surgical care. Expectations of surgical care emerged to be shaped by health literacy and perceptions of illness and surgery. Extensive information and health education campaigns on surgical conditions could improve healthcare seeking and empowerment of patients and their relatives. Reasons for seeking out alternative treatments can provide insights for future interventions. The care and treatment at large as well as the relationship with the healthcare worker impacted acceptability. Negative experiences by patients could be addressed by improving the training of medical personnel, enhancing interpersonal skills, and promoting a respectful environment in hospitals and healthcare settings. The needs of providers should be taken into consideration to improve motivation and workplace satisfaction. Structural improvements and an increase in qualified staff are the basis for a work environment in which acceptability can be given the necessary attention. Finally, our findings suggest the need for more extensive research on the underlying mechanisms of acceptability of surgical care, as an important foundation to tackle delays in surgical care seeking.

**Acknowledgements** We thank Global Health at University Hospital Bonn for the financial support, Ando-Modular aid e.V. for the inspiration for this study, Hatiika Namyalo for helping us navigate the Ugandan health care system and Grace Meara for helpful discussions.

**Contributors** PR, a German medical doctoral student at the time of the study, and WB, a professor for Global Health at the Universities of Cologne and later Bonn, developed the study design and the interview guides. Data collection was carried out collectively by PR and RN, a master's student at the School of Women and Gender Studies. PR then conducted the data analysis and prepared the manuscript. WB oversaw all aspects of this research. All authors contributed to the article and approved the submitted version. PR is the guarantor for the overall content of this article.

**Funding** This work was supported by the Open Access Publication Fund of the University of Bonn. The field work for this study was supported by the Section Global Health at the University Hospital Bonn, University of Bonn.

**Competing interests** None declared.

**Patient and public involvement** Patients and/or the public were not involved in the design, or conduct, or reporting, or dissemination plans of this research.

**Patient consent for publication** Not applicable.

**Ethics approval** This study involves human participants and was approved by Mildmay Uganda Research Ethics Committee under #REC REF 0102-2019. Participants gave informed consent to participate in the study before taking part.

**Provenance and peer review** Not commissioned; externally peer reviewed.

**Data availability statement** No data are available. We provided all relevant excerpts within the manuscript as raw data cannot be shared publicly due to the significant and valid confidentiality concerns of interviewees, small sample size and ethics approval requirements.

**ORCID iD**
Paula Rauschendorf http://orcid.org/0000-0003-1343-5516

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
