## [Reviewer comments · BMJ Open]

ARTICLE DETAILS

TITLE (PROVISIONAL)	Acceptability of surgical care in Uganda: A Qualitative Study on Users and Providers
AUTHORS	Rauschendorf, Paula; Nume, Rosette; Bruchhausen, Walter

VERSION 1 – REVIEW

REVIEWER	Schippert, Ana Carla Oslo Metropolitan University
REVIEW RETURNED	14-Feb-2023

GENERAL COMMENTS	The paper "Acceptability of Surgical Treatment in Uganda: A Qualitative Study on Users and Providers" examines a subject of vital importance to the care and health of surgical patients. Having worked in surgical services for many years and in multiple countries, I recognize the importance and significance of this study. This article, particularly the section on methods, needs additional clarity. I recommend applying scientific criteria. The manner in which the methodological framework is explained makes it look like it lacks rigor. Some of the references are more than 20 years old; if possible, I recommend replacing them. Several aspects must be addressed: Abstract Line 17, 18. I suggest: "Interviews were conducted with 32 past surgical patients, 16 community members who had not undergone surgery, 17 healthcare professionals involved in surgical treatment, and 7 District Health Officers or their deputies". Introduction According to the abstract results, the five intersecting categories that emerged were health literacy, perceptions, risks, and fears, search for alternatives, care/treatment, and trust in healthcare workers. In the introduction chapter, these aspects are not explained. My suggestion is to add a few lines connecting these factors to the acceptability of surgical care. The term health literacy should be defined, as well as what the authors of this study imply by the term. You have already gathered citations of studies in the discussion chapter that can be utilized to build a better introduction chapter. Line 40, 41, 42: I suggest: "This study was carried out to investigate patients' expectations
--

	and dissatisfactions with surgical care, as well as to understand underlying causes, by taking into account the perspectives of former patients, community members, healthcare professionals, and District Health Officers.” Methods Both grounded theory and theoretical saturation are not explained in this chapter in a manner that is clear and succinct. Additionally, the process of making the questionnaires that were used in the interviews should be described in detail here as well. A structure that is clearer and easier to follow would make this chapter significantly stronger. It would be beneficial for the article if there was a table that detailed the features of the sample (age, sex, etc.). Data analysis In this chapter, I believe it would be beneficial to provide a brief description of MAXQDA Pro Analytics. The findings present an accurate and comprehensive account of the topic at hand. I get the impression that there ought to be additional structure in the shape of quotes that are isolated from the remaining content in some way. As a direct result of this, I propose that the findings be presented in a manner that is distinct from what has been done. Citations in kursiv, for example Discussion On page 10, it says that there is evidence that "more health literacy has been proven to promote acceptability of surgical therapy for urine incontinence [20]". In this regard, I advocate for an expansion of this comprehension. It is somewhat limiting to solely discuss urine incontinence in this context while the topic of this study is more general. I recommend using literature that is more general in addition. For example: "Komenaka, Ian K., et al. "Health literacy assessment and patient satisfaction in surgical practice." Surgery 155.3 (2014): 374-383." Limitations In the section that is titled "Limitations," there is no mention made of the interviewer's possible effect in any other areas, as far as I can tell. The people who work in healthcare have a certain level of influence and professional authority, particularly in certain cultures and societies, and are seen as authorities. This is especially the case in countries where healthcare is not universally accessible. It's probable that this plays a role in perpetuating bias. I think it would be beneficial to highlight how/if the findings could be applied to different countries/ areas of healthcare.
--	---

REVIEWER	Nakku, Doreen Mbarara University of Science and Technology, ENT Surgery
-----------------	--

REVIEW RETURNED	26-Feb-2023
-------------

GENERAL COMMENTS	The authors have tackled a relevant area of study with findings that can influence service uptake. I commend them on expanding their study population to include the former patients and the community perceptions. below are key comments that require address:  1. The operational definition for acceptability is unclear. The authors imply that patients expectations and dissatisfactions as stated in the objectives is the definition for acceptability of surgical care rather than factors that influence acceptability. 2. Data collection: What role did data on access to the main road play in selection of community participants? How much time was considered between the time of surgery and selection of former patients? Could this have influenced the views of the participants on surgical experience and if so how does this affect the derived themes? 3. Results: The authors rightly acknowledged that the sample size was too small and not necessarily fully representative of the region and the country. In addition there is overlap of information in the derived themes. The results however are important in influencing future surgical service and planning. 4. Discussion: There is reference made to paediatric surgery care and access, however there is no indication of recruitment of paediatric participants. Did the team explore reasons why the former surgery patients do not follow through on the review visits to the hospital? This is important information that should be added if available.
--

VERSION 1 – AUTHOR RESPONSE

Reviewer: 1

Dr. Ana Carla Schippert, Oslo Metropolitan University, Akershus University Hospital

Comments to the Author:

The paper "Acceptability of Surgical Treatment in Uganda: A Qualitative Study on Users and Providers" examines a subject of vital importance to the care and health of surgical patients. Having worked in surgical services for many years and in multiple countries, I recognize the importance and significance of this study.

Response: Thank you very much!

This article, particularly the section on methods, needs additional clarity. I recommend applying scientific criteria. The manner in which the methodological framework is explained makes it look to lack rigor.

Response: The method section has been revised according to the suggestions that followed.

Some of the references are more than 20 years old; if possible, I recommend replacing them.

Response: All references older than 20 years have been removed or replaced, except for Luise White as we see her extensive work on rumours as a relevant reference for this study.

Abstract

Line 17. 18. I suggest: "Interviews were conducted with 32 past surgical patients, 16 community members who had not undergone surgery, 17 healthcare professionals involved in surgical treatment, and 7 District Health Officers or their deputies".

Response: This has been amended.

Introduction

According to the abstract results, the five intersecting categories that emerged were health literacy, perceptions, risks, and fears, search for alternatives, care/treatment, and trust in healthcare workers. In the introduction chapter, these aspects are not explained. My suggestion is to add a few lines connecting these factors to the acceptability of surgical care.

Response: The suggested connecting lines have been added.

The term health literacy should be defined, as well as what the authors of this study imply by the term.

Response: Health literacy is now defined in the introduction.

You have already gathered citations of studies in the discussion chapter that can be utilized to build a better introduction chapter.

Response: Additional references have been included to increase clarity of the introduction.

Line 40, 41, 42: I suggest: "This study was carried out to investigate patients' expectations and dissatisfactions with surgical care, as well as to understand underlying causes, by taking into account the perspectives of former patients, community members, healthcare professionals, and District Health Officers."

Response: The sentence was changed to fit the new flow of the introduction and to highlight the objective of the study

Methods

Both grounded theory and theoretical saturation are not explained in this chapter in a manner that is clear and succinct.

Response: Explanations were added to the method section.

Additionally, the process of making the questionnaires that were used in the interviews should be described in detail here as well.

Response: The process of creating the interview guide is now explained in depth in the method section.

A structure that is clearer and easier to follow would make this chapter significantly stronger.

Response: The structure of the method section was significantly revised to improve flow and clarity.

It would be beneficial for the article if there was a table that detailed the features of the sample (age, sex, etc.).

Response: Table 1 and Table 2 were added to provide demographic characteristics of the sample.

Data analysis

In this chapter, I believe it would be beneficial to provide a brief description of MAXQDA Pro Analytics.

Response: A description of MAXQDA was added.

The findings present an accurate and comprehensive account of the topic at hand. I get the impression that there ought to be additional structure in the shape of quotes that are isolated from the remaining content in some way. As a direct result of this, I propose that the findings be presented in a manner that is distinct from what has been done. Citations in kursiv, for example

Response: Citations were changed to cursive.

Discussion

On page 10, it says that there is evidence that "more health literacy has been proven to promote acceptability of surgical therapy for urine incontinence [20]". In this regard, I advocate for an expansion of this comprehension. It is somewhat limiting to solely discuss urine incontinence in this context while the topic of this study is more general. I recommend using literature that is more general in addition. For example: "Komenaka, Ian K., et al. "Health literacy assessment and patient satisfaction in surgical practice." *Surgery* 155.3 (2014): 374-383."

Response: Additional references on health literacy were added.

Limitations

In the section that is titled "Limitations," there is no mention made of the interviewer's possible effect in any other areas, as far as I can tell.

The people who work in healthcare have a certain level of influence and professional authority, particularly in certain cultures and societies, and are seen as authorities. This is especially the case in countries where healthcare is not universally accessible. It's probable that this plays a role in perpetuating bias.

Response: A section on possible observation bias was added.

I think it would be beneficial to highlight how/if the findings could be applied to different countries/ areas of healthcare.

Response: This was added to the limitations section.

Reviewer: 2

Dr. Doreen Nakku, Mbarara University of Science and Technology

Comments to the Author:

The authors have tackled a relevant area of study with findings that can influence service uptake. I commend them on expanding their study population to include the former patients and the community perceptions.

Response: Thank you very much!

below are key comments that require address:

1. The operational definition for acceptability is unclear. The authors imply that patients' expectations and dissatisfactions as stated in the objectives is the definition for acceptability of surgical care rather than factors that influence acceptability.

Response: A definition of acceptability has been added to the introduction and the article was revised for unclear statements regarding acceptability.

2. Data collection:

What role did data on access to the main road play in selection of community participants?

Response: Further detail has been added to the setting and participant's section.

How much time was considered between the time of surgery and selection of former patients? Could this have influenced the views of the participants on surgical experience and if so, how does this affect the derived themes?

Response: There were no exclusion criteria regarding the year of surgery. Table 1 shows that most patients had been operated within the 10 years before data collection. Unfortunately, some participants were not asked to provide the year of the surgical procedure. This has been added to the limitations.

3. Results: The authors rightly acknowledged that the sample size was too small and not necessarily fully representative of the region and the country. In addition, there is overlap of information in the derived themes. The results however are important in influencing future surgical service and planning.

Response: Thank you. Further discussion on the representativity was added to the limitations section.

4. Discussion: There is reference made to paediatric surgery care and access, however there is no indication of recruitment of paediatric participants.

Response: No pediatric patients were recruited. Paediatric surgery was used as an example due to scarcity of general data in some areas.

Did the team explore reasons why the former surgery patients do not follow through on the review visits to the hospital? This is important information that should be added if available.

Response: We agree that this would be valuable information. Unfortunately, this data is not available.

VERSION 2 – REVIEW

REVIEWER	Schipper, Ana Carla Oslo Metropolitan University
REVIEW RETURNED	18-May-2023

GENERAL COMMENTS	I'd like to take this opportunity to express my gratitude for the opportunity to provide feedback on such an interesting paper about such an important issue. I'd like to take this opportunity to express my heartfelt congratulations to the authors on their fruitful work in the form of the article review. Introduction The first draft of the article lacked a comprehensive introduction chapter; however, this section has since been improved, as have definitions for any terms that were unclear. The text flow has been improved and is now more fluid. Methods The section on methods is now clear utilizing scientific criteria, and demonstrates rigor. You write "The goal was..." on line 60 of the methods section. It would be better if you said something like "We conducted interviews with..." or "We interviewed..." Because you did, in fact, conduct interviews with the participants, this will be more accurate. You make a mention to "saturation" on page 5, but you don't provide a definition for it. Having a reference that provides information about "saturation" is also something that would be helpful. The findings are presented clearly. Discussion Line 42 on Page 12. It should be a focus, you write. It would be better to change it to "This should be a focus..." Line 48: A brief summary (just a few words) of the Fuller et al. and Dave et al. study (reference 29) would be advantageous. Good luck with your crucial research work! Final judgment: Accepted with minor modifications
--

VERSION 2 – AUTHOR RESPONSE

Reviewer: 1

Dr. Ana Carla Schippert, Oslo Metropolitan University, Akershus University Hospital

Comments to the Author:

I would like to express my appreciation for the opportunity to provide feedback on such an intriguing paper about such an important topic. In addition, I would like to take this opportunity to extend my sincere congratulations to the authors for their productive work in the form of the article review.

Response: Thank you very much for taking the time to review our research paper. We truly appreciate your valuable insights and positive feedback. We are grateful for your constructive suggestions, which have undoubtedly strengthened the quality and impact of the paper.

Introduction

The first draft of the article lacked a comprehensive introduction chapter; however, this section has since been improved, as have definitions for any terms that were unclear. The text flow has been improved and is now more fluid.

Methods

The section on methods is now clear utilizing scientific criteria and demonstrates rigor. You write "The goal was..." on line 60 of the methods section. It would be better if you said something like "We conducted interviews with..." or "We interviewed..." Because you did, in fact, conduct interviews with the participants, this will be more accurate.

Response: This has been amended accordingly.

You make a mention to "saturation" on page 5, but you don't provide a definition for it. Having a reference that provides information about "saturation" is also something that would be helpful.

Response: A definition of saturation and two references have been added.

Results

The findings are presented clearly.

Discussion

Line 42 on Page 12. It should be a focus, you write. It would be better to change it to "This should be a focus..."

Response: This has been amended accordingly.

Line 48: A brief summary (just a few words) of the Fuller et al. and Dave et al. study (reference 29) would be advantageous.

Response: A summary of the studies has been added